# Smart Arbitrary Waveform Generator with Digital Feedback Control for High-Voltage Electrochemistry

**Aleksey B. Rogov**

Scientific and Technical Centre "Pokrytie-A", Novosibirsk, 630015, Russia; alex-lab@bk.ru;
Tel +7-960-788-1729

**Abstract:** This paper describes a design approach to a control system of power supply for high-voltage electrochemical processes such as plasma electrolytic oxidation (PEO) or high-voltage anodising (HVA), which require alternating polarisation pulses up to 750 V and a typical current density of 50–500 mA/cm². Complex characteristics of the electrochemical system response on applied polarisations (positive or negative) cause necessity of precise control of polarising pulse shapes for better process operation and its understanding. A device performs cycle-by-cycle pulse-width modulation (PWM) control, including feedback based on digital analysis of the instantaneous current and/or voltage output, and the desired pulse waveform stored in memory for each output polarity. The output stage has four states corresponding to positive or negative pulses, as well as open- or short-circuit conditions, with respect to an electrochemical cell. A fully programmable controller allows one to generate arbitrary waveforms, as well as their sequences, by means of "regime designer" software. Moreover, a smart feedback system can provide adaptation of the next pulse parameter from analysis of the process prehistory. For instance, this approach allows one to separate main electrochemical process (coating formation) and diagnosis of the phenomenon through introduction of high-voltage triangular voltage sweep pulse within a pause of the main process, which is normally carried out under a current control.

**Keywords:** pulse shaping; plasma electrolytic oxidation; switching power supply; digital signal processing; digital feedback loop; proportional–integral–differential (PID); soft sparking mode

## 1. Introduction

High-voltage electrochemical processes such as plasma electrolytic oxidation (PEO), micro-arc oxidation (MAO), microplasma oxidation (MPO), or high-voltage anodising (HVA) allow one to form ceramic-like functional coatings on so-called valve metals (Mg, Al, Ti, Zr, Nb, Ta, etc.) [1–3]. They originated from conventional anodising by application of high-voltage polarisations (300–700 V) that caused local electrical breakdowns of the formed oxide film. The breakdowns (also known as "microdischarges"), accompanying the reaction of electrochemical substrate oxidation, transform amorphous deposited film into crystalline forms, as well as provide sintering and densification of the coating materials. Such coatings have perspective applications as antifriction and wear-resistive materials [4,5], anticorrosion protection [6–9], support for catalysts [10,11], bio-active [12], decorative [13–15], thermal barriers [16], electric insulating [17] layers, optical active coatings [18,19], magnetic shielding enclosures [20,21], etc. Unlike conventional anodising, a typical electrolyte solution for PEO does not include any mineral acids or heavy metals, it is normally a dilute solution (0.1–5 wt.%) of alkali, borate, aluminate, silicate, carbonate, phosphate, etc. that makes it ecologically friendly.

However, in spite of all the positive features mentioned above, PEO is still a rather exotic technology provided by a few companies with necessity of a deep "research and development" procedure for each particular part or component to be coated with PEO. This happens because of a lack of PEO process understanding. This is likely the most important limiting factor for widespread

PEO use in industry, rather than "energy consumption" mentioned by many researchers [22–27]. Typical energy consumption in PEO is between 0.05 and 15 kWh per square decimetre [28]; thus, readers can easily evaluate energy cost in any given region. However, a reduction in energy consumption is important in view of global energy saving and a reduction of harmful substance emissions.

The second important reason for restricted PEO application is a lack of suitable equipment, i.e., power supplies which match the requirements of PEO in the best way. This generates a "vicious circle"; on the one hand, without understanding, we are unable to design a suitable power supply, and, on the other hand, without a suitable power supply, we are unable to figure out the mechanism of the PEO process. Moreover, quite specific process requirements (hundreds of volts and tens of amps per square dm) strictly limit the list of available devices which are at least somehow suitable for PEO. As a result, the current approach, when experimental design is adopted to available equipment, dominates. This is likely a reason for stagnation in the commercialisation of PEO.

The devices working synchronously with mains frequency (circuitries based on thyristors [29], capacitors [30], or transformers [31,32]) were intentionally excluded from consideration, since they are unable to provide flexible output control, although they are suitable for some particular applications. All power supplies suitable for bipolar PEO can be divided into two main groups, operating under voltage or current control. Under voltage control, two independent direct current (DC) power sources are connected to the load through the output inverter that alternates load between the power sources [33–38]. Current control requires special approaches based on a special slope controller [39], inductive current sources [40], or current feedback [41,42].

Both voltage and current control modes allow one to extract additional information from current–voltage interrelation in respect of light emission [22,43–45], response of the system to a single step, analysis of relaxation [46–51], diagnosis of the electrochemical system by average values of short pulse series [52], etc. All these diagnostic approaches operate with rectangular pulses or pulses with sharp edges; therefore, it is extremely difficult to extract real current–voltage characteristics caused by the processes conjugated with coating formation, rather than recharging of internal or parasitic capacitance. In order to exclude (or minimise) capacitive response, a linear voltage sweep voltammetry can be applied. In such a case, the capacitive nature of the load will cause constant bias in current response, which linearly depends on the sweep rate, and it can be easily eliminated if necessary.

The aim of this work was to demonstrate a new approach, whereby a device was designed especially for studying the PEO process mechanism. Mixed polarisation conditions, including both current and voltage control mode, were applied where necessary. The design process included analysis of the typical PEO load behaviour, a description of the pulse shaping procedure and software, and experimental examination of the prototype.

## 2. Load Analysis

It is known that valve metals being immersed in a suitable electrolyte possess unipolar properties, i.e., current can easily pass when the metal is negative, but is suppressed when the metal is positive. Many years ago, this found practical application as an "aluminium rectifier" [53]. Currently, the blocking property of the positively charged valve metal electrodes is widely used in electrolytic capacitors [54]. From this, the metal-oxide–electrolyte junction can be considered as a special diode, which is open when the substrate metal is negative and closed under reverse polarisation. The main difference between this PEO diode and well-known electronic diodes (solid state or vacuum tube) consists of the presence of electrochemical reactions and corresponding mass transfer within the oxide layer accompanying coating growth.

Thus, the load for power supply in the PEO process is an electrochemical cell, which necessarily consists of a metallic working electrode (WE), covered with an oxide layer (OX) that is immersed in electrolyte solution (ES), which in turn is in contact with the counter electrode (CE) (see Figure 1a).

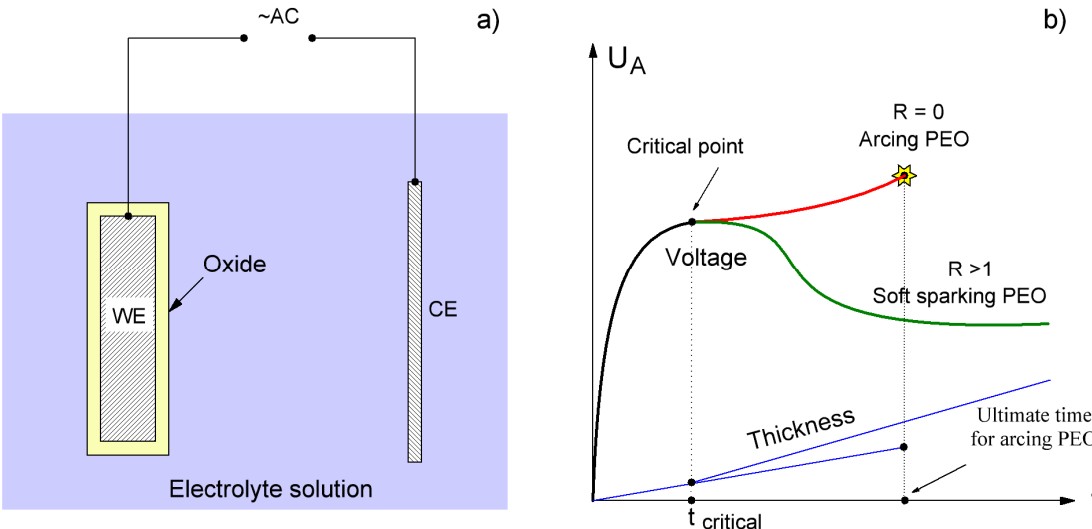

**Figure 1.** Schematic view of electrochemical cell (**a**) and evolution of anodic (positive) voltage peak (envelope of the positive pulses) and coating thickness during plasma electrolytic oxidation (PEO) processes of Al under current control (**b**). Value $R = J_n/J_p$ is the ratio of negative to positive current densities.

Both electrodes are typically made of metals and, thus, possess electronic conductivity. The oxide layer, depending on composition, structure, and morphology, may possess both ionic and/or electronic conductivity with a wide range of activation energies. The electrolyte solution is, normally, a pure ionic conductor (although sometimes molten salts can be used as well) [55]. In such a circuit composed of materials with various types of conductivity, the transfer of charges and masses includes overcoming potential barriers at interphase contacts, as well as migration and diffusion within bulk phases.

The ratio between average negative and positive current densities ($R = J_n/J_p$) is usually used as a generalised characteristic of the alternating current regimes. For instance, DC mode is characterised by $R = 0$, symmetrical alternating current (AC) mode is characterised by $R = 1.0$, an excess of positive current is characterised by $R < 1$, an excess of negative current is characterised by $R > 1$, etc. In addition, anodic and cathodic voltages represent peak values for the corresponding positive and negative pulse envelope. The importance of the peak voltages results from the fact that achievable electrical energy of the system is characterised by maximal applied potential difference, rather than by its average or effective (rms) values.

The growth of the coating is accompanied by transformations of the material and its morphology, thereby affecting the transport conditions that are usually observed as a variation of cell voltage under current control mode (Figure 1b). From the figure, it can be seen that in an arcing regime ($R = 0$ for Al alloys), both anodic voltage and coating thickness grow monotonically until ultimate conditions are reached; after that, the coating is affected by large arcs, which reduce the coating quality or destroy it. Under soft sparking conditions ($R > 1$ for Al), there is a "critical point" in the process, after which coating growth is accompanied by a decrease in anodic voltage. The latter is probably counterintuitive, since breakdown voltage is expected to increase with coating thickness; however, phenomena underlying the role of cathodic (negative) current are quite complex, and few attempts of its analysis were performed [56–58].

A deeper understanding of the load behaviour in PEO can be achieved with analysis of transient current–voltage curves (CVC) (Figure 2). For the very beginning of the PEO process, i.e., before the critical point (Figure 1b), there is no effect of cathodic current, and anodic CVCs have no hysteresis (Figure 2a, line $R = 0$). This CVC has a blocking region below the breakdown voltage ($U_{br}$), where strong capacitive response exists. This breakdown region of the CVC is characterised by high differential conductivity, and the shape of the curve is similar to the electrical breakdown of a zener diode (however, it is not the same). Once the critical point is reached with an adequate value of R

(Figure 1a), the process switches to soft sparking mode, and anodic voltage amplitude decreases; moreover, the hysteresis between upward and downward branches in the CVC can be observed (Figure 2a, $R > 1$). This hysteresis means that, after transition to soft sparking, the PEO coating is able to pass current at voltages much lower than the original breakdown level ($U_{br}$, Figure 2a). An additional feature of the hysteresis is a region with negative differential conductivity, which is located between low- and high-voltage conductive regions.

For negative (cathodic) polarisation, the thin PEO coating ($t < t_{critical}$) has responses with a nearly exponential growth of current similar to forward biased diode (Figure 2b). For negative polarisation, the transition to soft sparking is accompanied by an increase in the threshold voltage ($U_{TR}$) on the upward branch in the CVC. This is typically taken as a positive phase shift between voltage and current, which formally can be attributed to the inductive element in an equivalent schematic (Figure 3). It should be emphasised here that the inductive-like response is not attributed to real magnetic effects; hence, it is used only as a formal representation of observed behaviours in the equivalent circuit, which can be likely attributed to kinetics features of the charge and mass transfer (deposition, gas evolution) [51].

Thus, from an electronic engineering point of view, the PEO load can be considered as a high-voltage zener diode with a large reverse recovery charge and time (representing hysteresis in the anodic CVC) that depends on the specimen prehistory (level and duration of cathodic polarity).

Moreover, the coating growth makes this system essentially non-steady; therefore, the PEO processes at the very beginning (substrate metal covered with thin natural oxide) and at the end of coating formation (layer of a few hundred of microns) have a dramatic difference in electrical response that must be correctly operated by the power supply. A general description is given below.

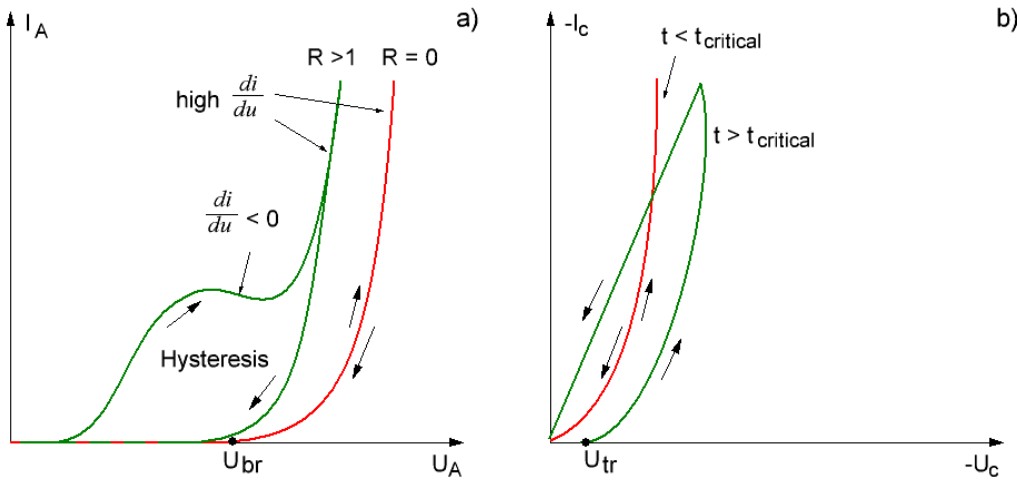

**Figure 2.** Transient current–voltage curves in PEO for anodic (**a**) and cathodic (**b**) polarisations. Symbols *t* and *R* have the same meanings as in Figure 1.

For positive (anodic) polarisation below the breakdown voltage, the PEO coating provides strong capacitive behaviour that is typically seen for a reverse biased diode. A simplified equivalent circuit is denoted in Figure 3 as a positive branch. Such a resistor–capacitor (RC) diagram is not surprising for a system where two conductors (metal and electrolyte solution) are separated with a poorly conducting oxide coating.

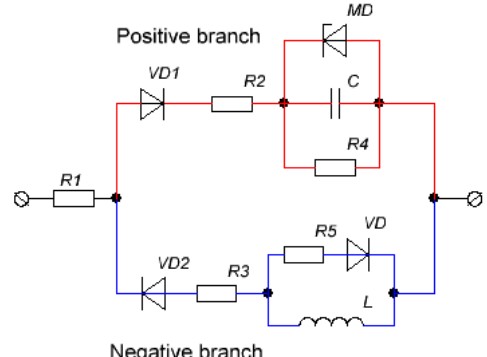

**Figure 3.** Simplified equivalent circuit of a PEO load for positive and negative polarity, based on data from Reference [50]. MD—zener diode of microdischarges; R1—resistance of electrolyte; R2,R3—resistances of the coating for positive and negative current; R4—differential resistance of microdischarges; R5—resistance characterising self-decay of the negative induced states; C—capacitance of metal-oxide–electrolyte junction under reveres bias; L—virtual inductance attributed to slow chemical processes (absorption, desorption, surface diffusion, etc.); ideal diodes VD1,2 formally separate positive and negative branches..

Furthermore, this simple schematic is consistent with the results of small-signal electrochemical impedance spectroscopy (EIS) [59–61], as well as transient analysis [50,51], of current and voltage pulses at different frequencies [46,48].

It is obvious that transient processes for a capacitive load, which is connected to a voltage source, are rather complex and the current spike is limited only by electrolyte resistance and inductivity of the wires. Those spikes often require extra power capabilities of employed electronic components. They also reduce reproducibility of the final coating. Moreover, such spikes are probably able to ignite some breakdowns at specific conditions that affect coating continuity and corrosion resistance. To eliminate the spikes, the pulse has to be formed either under current control or under limited voltage rise time. Inductive-like behaviour under negative polarisation means that passage of certain charge requires the application of a certain voltage for a given time (similarly to the V·s product in real magnetic phenomena). Since PEO is an electrochemical process and the rate of electrochemical reactions is directly proportional to the number of passed charges, the current control mode looks like the best choice for PEO processes both for negative and positive polarities.

However, voltage control or combined "current control with limited voltage" modes of operation could be useful, for instance, in PEO on complex-shaped substrates. Moreover, voltage control is required for CVC analysis or voltammetry, since CVC may include regions with negative dI/dU differential resistance (see Figure 2). Therefore, the final design of the power supply for high-voltage electrochemical processes should be able to provide both current and voltage operational control.

## 3. System Design

### 3.1. Waveform Shape Generation

The desired output waveform is stored in the memory as a list of set points. Each set point $n$ is described in terms of output magnitude ($A_n$) at a given particular time ($t_n$), state of output inverter (positive, negative, open, closed), and control function $y = f(u,i)$. This later allows us to choose which output parameter is employed for feedback and set point values. Normally, $y = u$ for the voltage control mode and $y = i$ for the current control mode, where $u$ and $i$ are the values attributed to instantaneous output voltage and current, respectively. However, if necessary, more complex functions can be employed; for instance, the function $y = i*u$ corresponds to power control mode, or the function $y = a \times i + b \times u$ (where a and b are coefficients) may be used for the design of the specific

output current–voltage characteristic. In addition, some additional information can be associated to each set point such as an external trigger control or a start position of the introducing diagnostic pulse. Moreover, the set point list includes headers, which define the number of pulse repetitions, as well as the end flag and service data. The structure of set points, their list, and the corresponding discreet functions are illustrated in Figure 4.

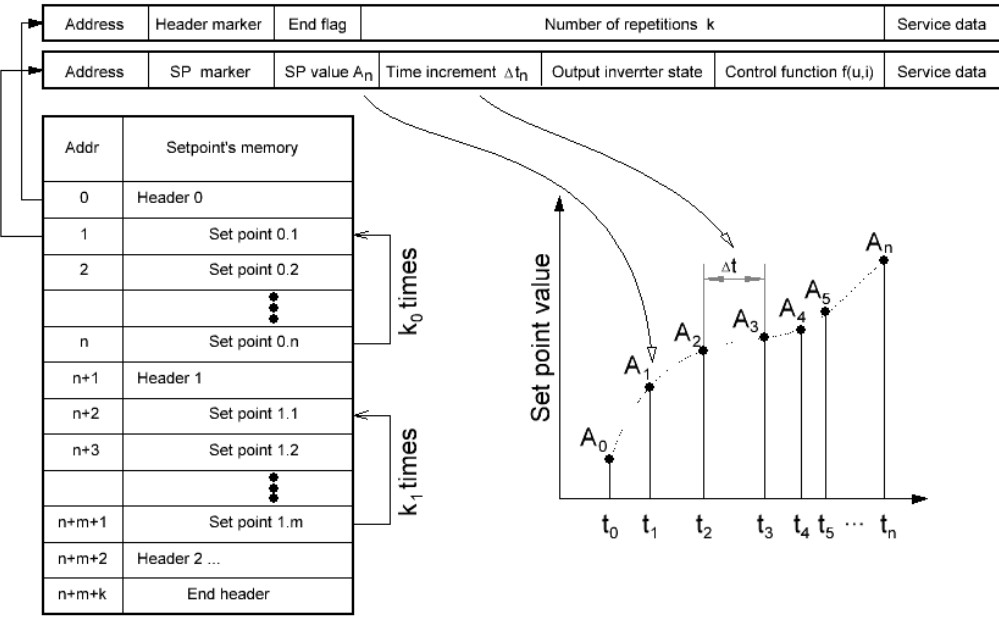

**Figure 4.** Illustration of the list and structure of set points and their corresponding waveform.

Interpolation between two set point values occurs on the fly with particular values of the intermediate ($k_{th}$) set point calculated as follows:

$$A_{n,k} = A_{n-1} + k \times \tau \times (A_n - A_{n-1})/(t_n - t_{n-1}), \tag{1}$$

where $\tau$ is a period of pulse-width modulation (PWM) frequency.

A diagram illustrating the phase shift technique is depicted in Figure 5. The counter register (CNT) continuously increases its value from 0 to a fixed maximum level ($PWM_{max}$), while both complimentary output units (PWM1, PWM2) toggle from a high to a low state and vice versa, when the CNT approaches corresponding comparison levels (PWM_ARR1, PWM_ARR2). Channel PWM1 operates in steady-state conditions; therefore, PWM_ARR1 is always equal to zero. In order to achieve a phase shift in channel PWM2 with respect to PWM1, the PWM_ARR2 can be changed between zero (0° shift) and $PWM_{max}$ (180° shift). Since the output signal $PWM_{out}$ is proportional to the area overlapped by PWM1 and PWM2 signals, a 0–180° shift corresponds 100–0% of the output. The relationship between the value of PWM_ARR2 and the desired output duty ratio is PWM_ARR2 = $PWM_{max} - PWM_k$.

The device operates in discrete mode, analogue-to-digital conversions (ADC) are synchronised with periods of PWM signal, i.e., the feedback loop can affect only the next period of pulse-width modulated signal, driving the power module. On one hand, this deficit could be minimised with an increase in PWM carrier frequency. On the other hand, the minimum PWM period is limited by the calculation time required by digital feedback control (see Figure 5). Within each particular working period, the calculations should be done and the PWM_ARR2 register should be updated before the CNT register reaches $PWM_{max}$, when the PWM timer updates its working registers.

The adjustment of the output signal in accordance with set points values is carried out by a proportional–integral–differential (PID) controller in the form of difference equation carried out by an Advanced RISC Machines (ARM) core.

$$PWM_k = PWM_{k-1} + K_p(e_k - e_{k-1}) + \frac{1}{K_i} \times e_k + K_d \times (e_k - 2e_{k-1} + e_{k-2}), \tag{2}$$

where $PWM_k$ is the duty ratio for step $k$, $e_k = A_{n,k} - f(u,i)_k$ is the difference between set point value and output control function, and $K_p$, $K_i$, and $K_d$ are coefficients for the proportional, integral, and differential terms, respectively. Normally, latter PID coefficients are not once defined as constants since optimal signal parameters can be achieved only taking into account specific properties of the load (at least positive and negative waves should have a separate set of PID coefficients) and pulse parameters, such as rise and fall times, allowance of overshot, etc. An employment of fuzzy logic control could bring additional profit in the accuracy of the output signal.

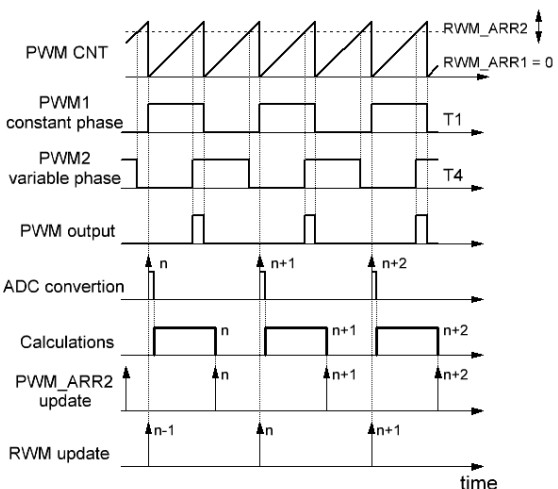

**Figure 5.** Phase shift pulse-width modulation (PWM) timing diagram. Complementary signals and dead times are not shown.

### 3.2. Correction of the Long-Term Inaccuracies

Since pulse shaping is carried out within a limited time, a considerable systematic relative error in output pulse amplitude will often take place, normally from 10% to 50% depending on the particular pulse shape. Such inaccuracy can be avoided through an application of a secondary stabilisation loop, which compares the actual average or peak values with those derived from a set point list. If discrepancy exists, each set point value is multiplied by a correction coefficient using the secondary proportional–integral (PI) controller until desired values are achieved. A block diagram of the correction procedure is shown in Figure 6. An example of how the system works is shown below. Once measurements of passed charges ($Q_m$) are completed, the difference $\Delta$ can be calculated as follows:

$$\Delta = Q_0 - Q_m, \tag{3}$$

where $Q_0$ is the desired average value. This error in accordance with the proportional–integral approach is transformed to a correction function.

$$\gamma = Kp_2 \times \Delta + {}^1/_{Ki_{2_2}} \times \textstyle\sum \Delta, \tag{4}$$

Then, this function can be transformed into a scaling coefficient, $\beta$.

$$\beta = \gamma / Q_0. \tag{5}$$

Finally, the input set point value for PID controller (see Section 3.1) appears as

$$A_{nk}^* = (1 + \beta) \times A_{nk}, \tag{6}$$

thereby sustaining the correct average value of the output current.

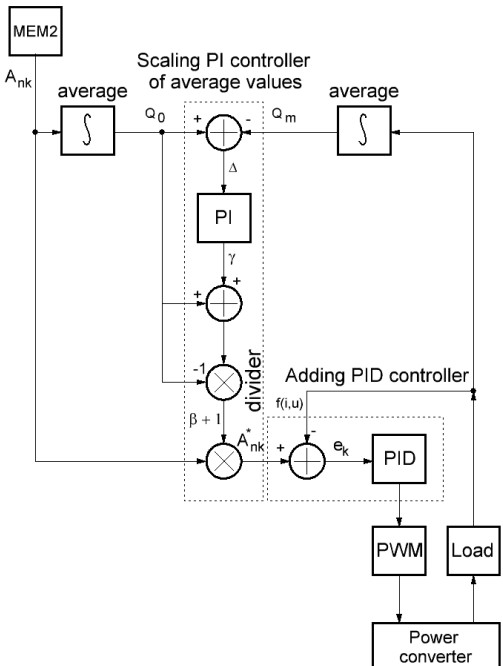

**Figure 6.** Block diagram for pulse shaping and the correction of systematic inaccuracies. MEM is the set point memory, and PWM is the timer with PWM output.

### 3.3. Prototype Details

An experimental investigation of the proposed approach to pulse shape generation on a PEO load was performed with a laboratory-scale prototype including a power unit, personal computer (PC)-based software, and a electrochemical cell.

The power unit transformed the mains electricity into an output signal under PWM control. The power unit's particular design depended on the relationship between input voltage and desired output levels. For example, a step-up converter is required for mains with 230 V$_{AC}$ if a PEO process of Al alloys is necessary, since typical breakdown voltage is in the range of 350–650 V. In contrast, if mains provide 440 V$_{AC}$ voltage and Ti alloys are to be coated, a step-down converter is enough, since the PEO of Ti runs typically within the 200–400 V range. The requirements for the power unit include a suitable performance for generation of desired rise/fall times. Another important feature of the developed power stage is that a few units are able to operate on the same load in parallel by connecting together appropriate terminals (A1, A2) and synchronous control.

Since, in our experiments, output power was limited to around 2 kW, this device was powered by single-phase mains. Therefore, a step-up converter based on full-bridge topology was employed (Figure 7a, T1–T4). Auxiliary blocks such as an input filter, a power factor corrector, an input rectifier, and general protections are expected as included where necessary without further discussion. A simplified diagram of the pulse shape forming is depicted in Figure 7b.

A power converter operated under phase shift PWM control is shown in Figure 7(T1–T4). As a driver for power switches (T1–T4), conventional optical, capacitive, or transformer-insulated approaches can be employed; however, for the output stage, only drivers which are capable with DC operation are required (optical, capacitive-coupled). L1C1 and L3C3 are the input and output of the low-pass filters, respectively. L2 is an inductor with a saturated core, C2 is the capacitor removing the DC bias of the transformer (Tr), the RC circuit reduces parasitic oscillation spikes, VD is the full-bridge rectifier, R1, R2 is the voltage divider, R3 is the low inductive current shunt, T1–T9 are the IGBT (or MOSFET for low input voltage) transistors, and the load is the electrochemical cell, including a working electrode (WE) immersed into an electrolyte solution and a counter electrode (CE) (see Section 2).

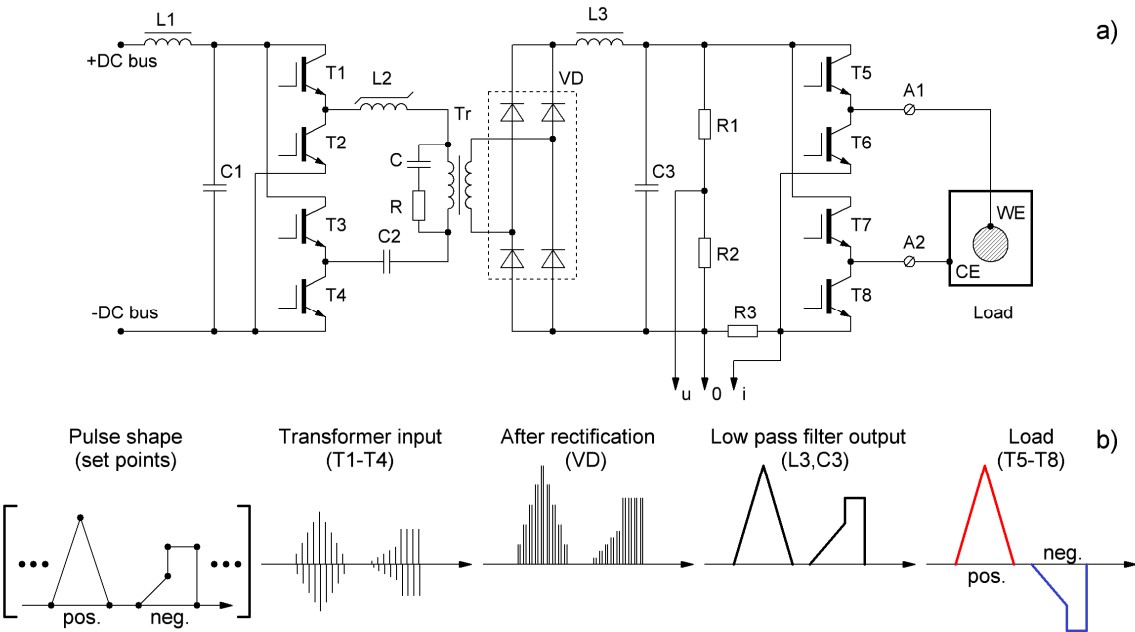

**Figure 7.** Schematic diagram of the double full-bridge power unit (**a**); simplified path of pulse shape from memory to the load (**b**).

The output stage of the power supply includes four switches (T5–T8) connected as a full-bridge schematic. Each particular switch is controlled independently. As a result, four various connections can be performed between the power unit and the load. Firstly, the load can be connected to the power unit with positive (anodic) or negative (cathodic) polarities. Secondly, within pause, the load can be in a state of open or short circuit for discharging of the remaining charges. Moreover, under normal application conditions, all switching within the output stage occurs at zero input current from the power converter, providing safe soft switching. Typical delay between PWM being off and the output stage switching is one PWM period. However, in special situations (e.g., metal–electrolyte combination with a high dissolution rate), hard switching of the output stage is also available. The latter allows one to more easily initiate the breakdowns using impact high-voltage excitation.

Instantaneous measurements of the output current and voltage are performed with resistive sensors (R1–R3). In such a case, the microcontroller unint (MCU) turns out under the potential of the high-voltage side, and communication occurs via an optically isolated interface between the control PC unit and secondary power units, when parallel operation is necessary.

The prototype had the following specifications:

- (a) Output voltage (max), V: ±750;
- (b) Output current (max), mA: ±2500;
- (c) Current rise/fall time (max), μs: 100;
- (d) Input: 230 V ±10%, 50 Hz;
- (e) Operation modes: DC, unipolar pulses, bipolar pulses;
- (f) Communication: RS232, USB, WiFi;
- (g) PWM frequency, kHz: 50,000;
- (h) ADC resolution (effective): 12 (10);
- (i) Protections: safety interlock, over current, over voltage, overheating;
- (j) Cooling: active airflow;
- (k) Size/weight (mm/kg): 400 × 270 × 170/5;
- (l) Accuracy (waveform/average): better than 5.0/1.0%;
- (m) Topology: phase shift full bridge.

The main components and design features are listed below.

- (a) Power switches: IGBT IKW120N15;
- (b) Isolated drivers: HCNW3120;
- (c) Transformer core: ETD 59;
- (d) Output inductor core: ETD 59 (gap 1.5 mm);
- (e) MCU model: STM32F405;
- (f) Primary safety control: current transformer;
- (g) Secondary monitoring: resistive shunt, resistive voltage divider;
- (h) WiFi module: ESP8266;
- (i) Isolated UART/USB bridge: 6N137 + CH340.

Because of the complex structure of the output signal, it is convenient to use "regime designer" (Figure 8) software, which allows the operator to synthesise the most often used output waveforms and their sequences as pulse trains and a set point list ready for the processing. Typically, the basic regime designer includes various combinations of trapezoidal waveforms defined by four points for each polarity (see Figure 9). As a result, triangular, saw-tooth, and trapezoidal waveforms can be easily generated. However, if necessary, a more complex waveform can be loaded from an external tabulated file. Moreover, regime designer allows one to include special diagnostic sequences, as well as markers, for external triggers that allow synchronising outstanding oscilloscopes or camera with particular process moment.

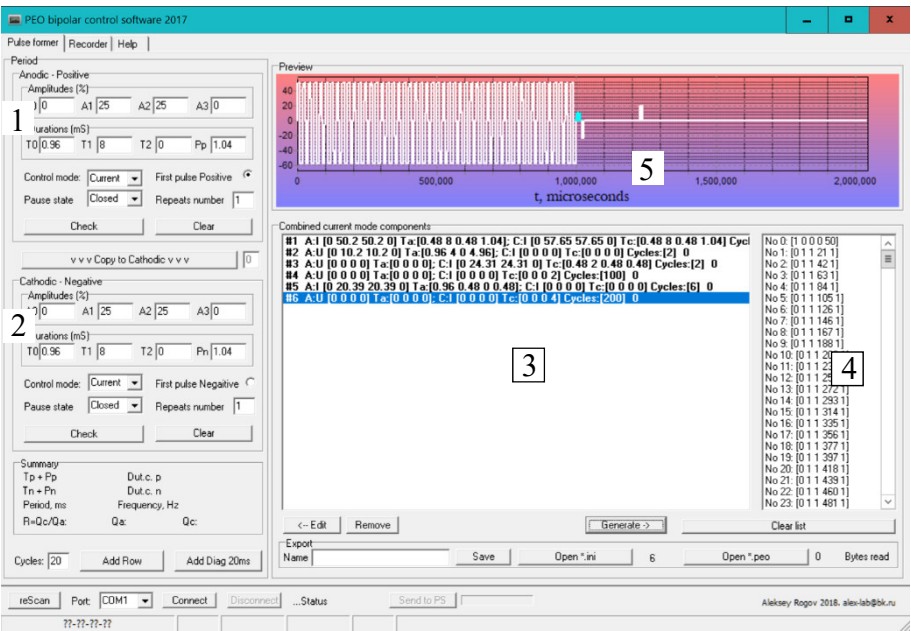

**Figure 8.** Appearance of regime designer software. Legend: 1, 2—parameters of anodic and cathodic pulses, respectively; 3—list of regime elements for editing; 4—list of set points; 5—simplified preview of the waveform.

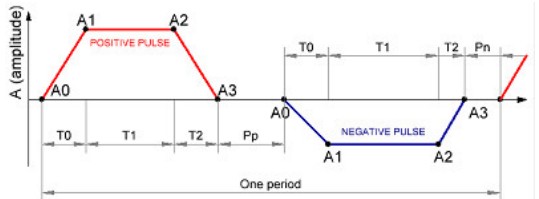

**Figure 9.** Sketch of the trapezoidal pulse waveform and corresponding parameters.

In addition to regime designer, the software part of this prototype includes monitoring facilities, which acquire data from the device, provide preliminary calculations, plot them both as time-chart graphs and as waveforms, and save data to a storage for further analysis (Figure 10). All communications between the power supply and the controlling computer are performed via either isolated USB or Wi-Fi interfaces.

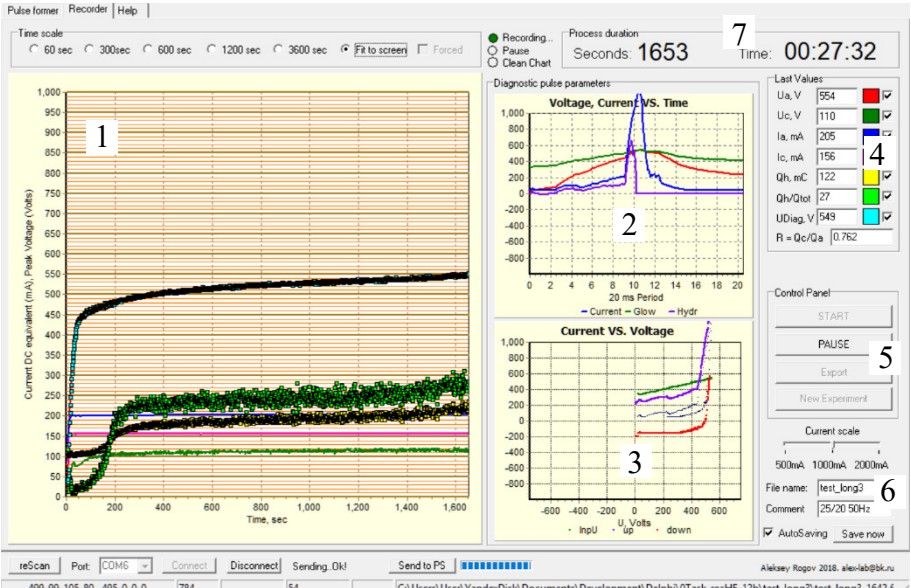

**Figure10.** Appearance of monitoring software. Legend: 1—process time chart of anodic and cathodic voltages and currents, as well as additional process parameters calculated from this data; 2—diagnostic pulse waveform; 3—*XY* graph of diagnostic pulse as current–voltage characteristics; 4—numerical values of measured parameters (Ua,Uc—anodic and cathodic voltages; Ia,Ic—anodic and cathodic currents; Qh—absolute value of asymmetric charge (see below); Qh/Qtot—percentage of asymmetric charge with respect to the total charge within the diagnostic pulse); 5—process controls; 6—waveform auto save settings; 7—timer.

## 4. Results and Discussion

As an example of the device performance, four different functions (I–IV) were tabulated as anodic pulse shapes, whereas cathodic pulses were simple rectangular (Figure 11a). With Microsoft Excel, a table was exported to a CSV file and uploaded into the power supply with twofold repetition of each pulse in the series. Experimental waveforms of load voltage and current are depicted in Figure 11b.

In view of intelligent process control, the combined polarisation conditions, including producing the pulse train (Figure 11c) under current control and a single diagnostic pulse under voltage control, can be considered (Figure 11d). In addition to mixed operation (current/voltage), this regime is complicated by the fact that diagnostic pulse amplitude (in terms of voltage) cannot be taken as an initial parameter, since breakdown voltage evolves during coating growth.

Thus, in order to provide reasonable diagnostic conditions, i.e., conditions that match particular coating properties (breakdown voltage), the diagnostic pulse amplitude has to be adopted in accordance with the maximum voltage within a prior producing pulse train. In other words, after the end of the producing pulse train, a special diagnostic procedure should be carried out that includes (a) finding the maximum voltage, (b) recalculating the function of the diagnostic pulse, and (c) sending the acquired data to the PC for further analysis. Figure 11d illustrates relations between the voltage envelope (U$_A$) during the producing pulse train and amplitude of diagnostic pulse (marked by arrow).

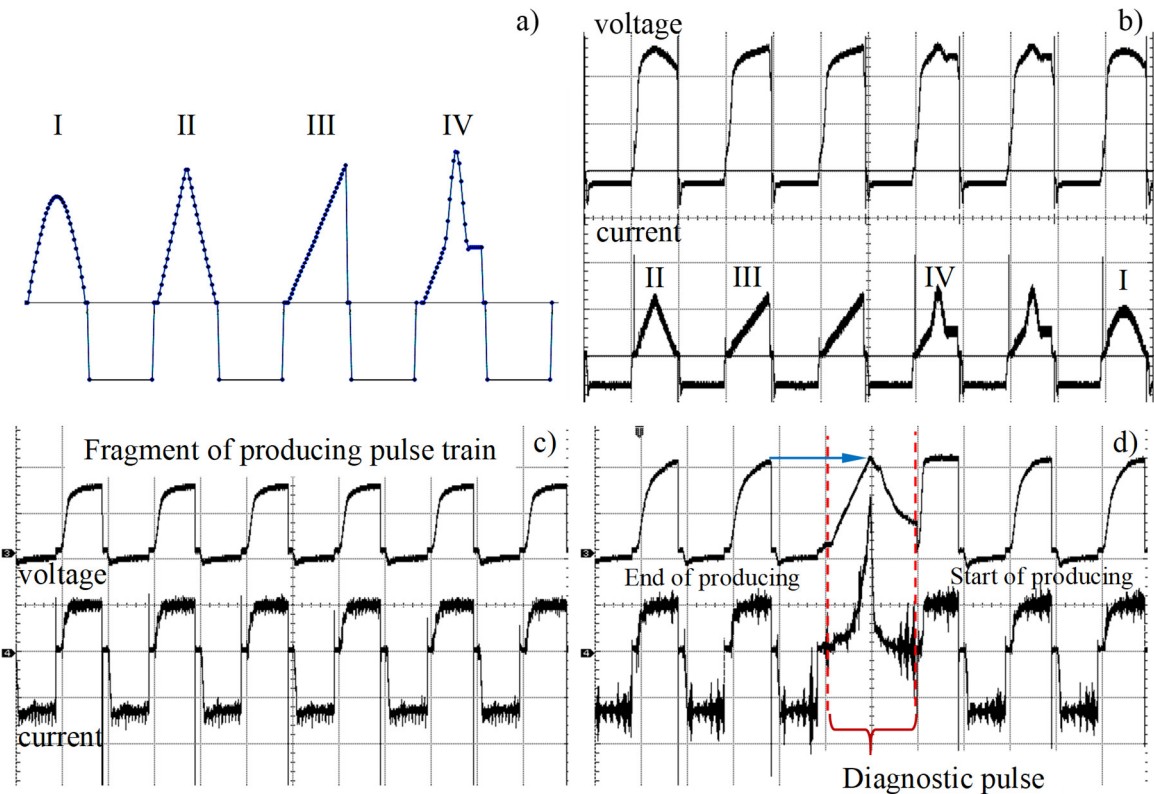

**Figure 11.** Scheme of tabulated set points for different functions of current (**a**): I—sinusoidal half wave, II—triangle, III—saw tooth, IV—function sequentially composed of linear sweep, part of the sinusoidal, and a constant value; (**b**) corresponding voltage and current output waveforms, as well as output signal waveform within (**c**) the producing pulse train, and (**d**) the triangle diagnostic pulse. PEO process on A2024 Al alloy in a silicate–alkaline electrolyte; 250 V/div, 500 mA/div, 10 ms/div.

An example of a diagnostic pulse response is depicted in Figure 12a. The green area represents the asymmetrical part of the current, which is proportional to the hysteresis in the CVC (see Figure 2a), and the $Q_h$ symbol corresponds to the absolute value of charge (coulombs) represented by this area (Figure 12, Iup–Idown). In this test, the additional ADC input was used for acquisition of the intensity of light (L, a.u.) produced by microdischarges (Figure 12b) with the BPW21 photodiode loaded to a 10k resistor.

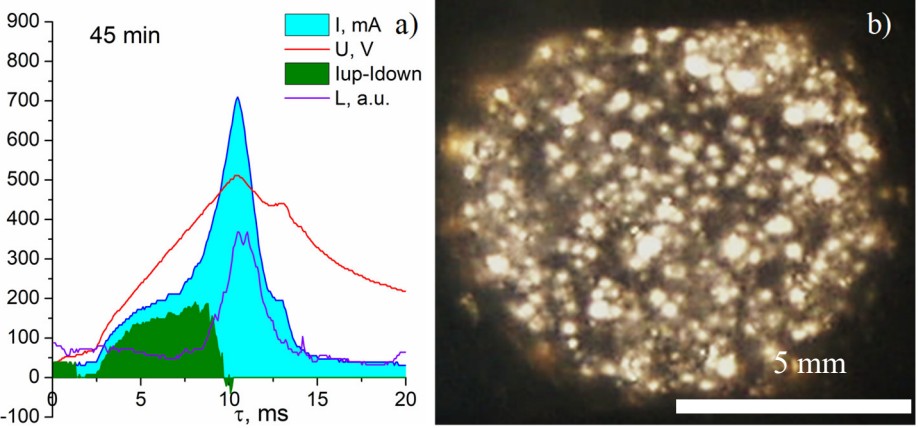

**Figure 12.** Example of a diagnostic pulse response (**a**) and appearance of the microdischarges (**b**) in course of the PEO process of Al in a silicate–alkaline electrolyte. Legend: I, mA—current; U, V—

voltage, Iup–Idown—difference between upward and downward branches of current; L, a.u.—photodiode current.

Figure 13a illustrates the behaviour of the anodic voltage amplitude and the corresponding diagnostic pulse value during the first 2 min of the process. It can be seen that the auto adjusting system caught the correct voltage level by the first minute; after that, through the whole process, the average discrepancy between the anodic voltage amplitude (U+) and the diagnostic pulse maximum value ($U_{diag}$) was around +3 V (Figure 13a inset), except for the period of transition to the soft sparking mode (~29 min), where average discrepancy achieved a level of −8 V.

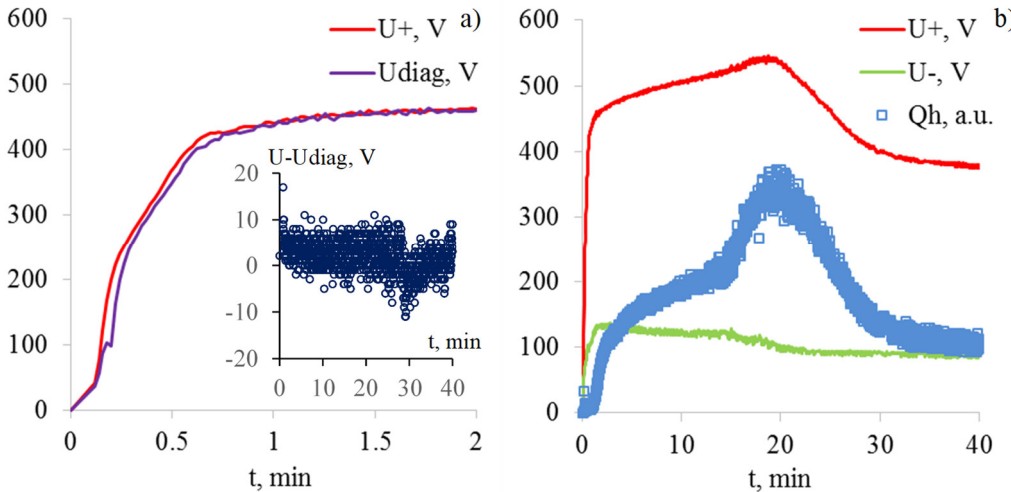

**Figure 13**. Initial period of anodic voltage (U+) of PEO process under current control, producing a pulse train and a self-tuning diagnostic pulse amplitude (Udiag) (**a**). The inset illustrates the discrepancy U–Udiag within the entire process. Evolution of the positive and negative voltage amplitudes and the hysteresis charge (**b**).

Finally, Figure 13b demonstrates the evolution of the main electrical parameters of the PEO process (U+, U−), as well as characteristic function Qh, proportional to the charge associated with hysteresis in the anodic CVC, thereby establishing the numerical parameter connecting the associated processes under negative and positive polarisations in course of the PEO process. It is expected that this parameter helps disclose the internal mechanism of PEO, and improves the efficiency and reproducibility of the coatings. Finally, the advanced control method can be employed, when control function $y = f(u,i)$ (see Section 3.1) includes additional parameters (and/or conditions) derived from analysis of the diagnostic pulse response, optical signal, etc.

## 5. Summary

The device described in this work allows one to carry out high-voltage electrochemical processes, as well as analyse charge and mass transfer in situ. Flexible control of the operational mode (current/voltage controls), pulse shape, and polarity, as well as instantaneous data acquisition with fast on-board analysis will bring high-voltage electrochemistry to a new level of understanding and control due to intelligent adjustment of the local process parameters with the help of digital feedback.

A control function term associated with the external input can be connected to various sources of process information (optical signal, spectrometer, thermometer placed in either solution or substrate, acoustic microphone, etc.), allowing the coating formation process to be comprehensively controlled. A further development of the presented approach includes an introduction of perturbation into the output signal for frequency response analysis of the load in course of the coating formation process.

It is expected that a comprehensive understanding of the process basis will considerably extend the applicability of this electrochemical technique to a wider range of applications, where reproducibility, energy efficiency, and ecology compatibility are the main important features.

**Acknowledgments:** The author would like to express his very great appreciation to Andrey Belousov for productive discussion.

**Funding:** This research received no external funding

**Conflicts of Interest:** The authors declare no conflicts of interest.

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
