# Peer review of "Smart Arbitrary Waveform Generator with Digital Feedback Control for High-Voltage Electrochemistry"

_instruments, doi:10.3390/instruments3010013_

Reviewer 1 Report

I recommend changing the word "injection" for the diagnostic pulse to "introduction".

Lines 226, 230, extra commas in the starting positions

Fig. 7. I do not see the necessity of drawing the first stage of the bridge. You indicate +DC and -DC buses before L1. The same holds for the lines after L3. We see no 220/380 mains, rectifier and filter drawn in the input of the schematic, so it is incomplete anyway. Please redraw the circuit starting from L3 and correct the explanation.

Lines 50, 56 and so on. Having one author and writing "We" looks strange. Are you sure that all the authors appear in the list? This needs correction.

Author Response

Dear Reviewers and Editor,

Thank you very much for useful comments, correction and productive discussion. Please find below my answers.

Best regards,

Aleksey Rogov

Reviewer #1.

Q1. I recommend changing the word "injection" for the diagnostic pulse to "introduction".

A1. “Injection” has been replaced by “introduction”.

Q2. Lines 226, 230, extra commas in the starting positions

A2. Commas were removed.

Q3. Fig. 7. I do not see the necessity of drawing the first stage of the bridge. You indicate +DC and -DC buses before L1. The same holds for the lines after L3. We see no 220/380 mains, rectifier and filter drawn in the input of the schematic, so it is incomplete anyway. Please redraw the circuit starting from L3 and correct the explanation.

A3. In order to clarify operational procedure Fig.7b has been added.

The mains was not shown due to various sources can be used to power this device. First bridge (T1-T4) is essential part of the schematic since output signal modulation (in absolute scale) happens at this stage by PWM (50kHz). Filter L3,C3 is a low pass filter, which removes carrier frequency of the first stage PWM (above 10kHz), but envelope of of the signal, which shape was programmed, is fed to output stage. Second bridge (T5-T8) is required only for polarity switching of the output.

Q4. Lines 50, 56 and so on. Having one author and writing "We" looks strange. Are you sure that all the authors appear in the list? This needs correction.

In line 50 and 51, “we” means author and readers. However, in other places, it was used incorrectly and phrases had been rearranged.

Reviewer 2 Report

The paper is interesting, but I would like toto see the SEM and confocal photos of obtained surfaces after those PEO treatments. However, in my opinion, in this form to that journal is ready for printing. 

Author Response

Dear Reviewers and Editor,

Thank you very much for useful comments, correction and productive discussion. Please find below my answers.

Best regards,

Aleksey Rogov

Reviewer #2.

Q1. The paper is interesting, but I would like toto see the SEM and confocal photos of obtained surfaces after those PEO treatments. However, in my opinion, in this form to that journal is ready for printing. 

A1. In this paper, I have  intentionally concentrated attention to details of the power supply design, avoiding any material characterisation and mechanism of the process as it is out of scope of the “Instruments” journal. Details of coating formation process will be published later.

Reviewer 3 Report

The article submitted for review in the journal is a compelling proposition describing a power supply suitable for plasma electrolytic oxidation (PEO) research of valve metals (principally, alloys of aluminum and titanium). The author has carried out a thorough literature survey and possesses a solid knowledge of the state of the art of PEO treatment, and the problems that plague the commercialization of the technique. I am no conventional to the power electronics and the design of the high-voltage power supplies, as well as the ones that are suitable for the PEO process. However, from the point of an electrochemist that is actively researching in the subject, I am in the position that could improve the clarity of the article to a broader audience that could greatly benefit from it. I strongly agree with the author that there exists a sort of an impasse between the development of novel power supplies that meet the strict requirements of the PEO coating setup and the deep understanding of the process itself. The paper I was presented with seems to attempt to close that gap; therefore, I am actively opting in favor of its publication.

Nevertheless, in its present form, the manuscript is not flawless which is why I would be grateful if the author could address some of my recommendations, comments, and questions which I upend below.

1)      English errors and editing should be improved for the sake of clarity and comfortable reading. Some examples of their misuse are the following:

a.       Page 1; line 22: “(…) which normally carried out under a current control.”
It should be “(…) which IS normally carried out under a current control.”

b.       Page 2; line 50: “(…) “vicious circle”: on one hand, without understanding we are unable to design suitable power supply, (…)”.
It should be “(…) “vicious circle”: on THE one hand, without understanding we are unable to design suitable power supply, (…)”.

c.       Page 2; line 52: “(…) hundreds of Volts and tens of Amps per (…)”
Units should not be written with a capital letter.

d.       Page 2; line 72: “ (…) and it can be easily eliminate if necessary.”
It should be “and it can be easily eliminateD if necessary.”

e.       In multiple instances in the text, the author misspelled the word “envelope” by writing it in the verb form (envelop).

f.        Page 3 & 4; lines 121 & 128. The use of sub- and superscripts in the manuscript is, in general, in order. The only examples of not following the routine was in the case of U_br, in the two quoted lines.

g.       Page 9; line 281. The author has written “us” meaning “microseconds”. Please use the corresponding Greek letter “µ”.

h.       Reference to Fig.9 precedes that of Fig.8, which should not happen. Consider changing the order of figures or edit text to follow this convention.

More examples can be found in the text. The style adopted by the author does not render the article unreadable but a second look by a native speaker would considerably benefit the article.

2)      In Fig.2. There seems to be a slight mistake in the notation of time (t). In the caption, there is an annotation “Symbols t and R have the same meaning as in Fig.1.” It is slightly misleading because in Fig.1. the author used a non-abbreviated description of the x-axis, namely, “time” and not “t”.

3)      As I’ve mentioned, I am not of an electronics/electrical engineering background. Therefore, it would be the most welcome to elaborate briefly in the text on the meaning of Fig.5, and what does this signal processing procedure correspond to concerning the optimal work of the power supply during PEO coating formation.

4)      It feels wrong to have “4.1 Prototype details” subsection in “Results and discussion”. Please consider moving it to the previous section. Alternatively, you may find it more fitting to make it a separate section of an article altogether.

5)      Please clarify what is meant in the sentence (Page 10; line 308): “Moreover, regime designer allows one to include special diagnostic sequences as well as markers for external triggers that allow to synchronise outstanding oscilloscopes with particular process moment.”

6)      In Fig.10., please clarify:

a.       What is the meaning of Q_h and Q_h/Q_tot. This quantity is also brought up in Fig.13b.

b.       I am not sure what are the meanings behind legends of charts 2 and 3 (Current, Glow, Hydr; and InpU, up, down).

7)      In Fig.11., please try to indicate that L corresponds to the light intensity of the MDs (at least I suspect that is the reason behind it). I would also like to know, is collecting this additional ADC data trigger some response from the power supply that modifies the output signal to the process? Simply put, do the light emission observations affect how the device runs the process?

8)      As a follow up to the previous question, I would like to know, because it is not immediately clear from the article: does the result of the diagnostic pulse affect the process conditions in the subsequent pulse train? It would be valuable to use such a diagnostic pulse for the sake of studying the process or the reproducibility of the desired quality of the coatings. However, being able to tune the power supply to respond specifically to the given diagnostic pulse response would be an outstanding step forward in the commercialization of PEO in industry.

Author Response

Dear Reviewers and Editor,

Thank you very much for useful comments, correction and productive discussion. Please find below my answers.

Best regards,

Aleksey Rogov

Reviewer #3.

1)      English errors and editing should be improved for the sake of clarity and comfortable reading. Some examples of their misuse are the following:

a.       Page 1; line 22: “(…) which normally carried out under a current control.”
It should be “(…) which IS normally carried out under a current control.”

b.       Page 2; line 50: “(…) “vicious circle”: on one hand, without understanding we are unable to design suitable power supply, (…)”.
It should be “(…) “vicious circle”: on THE one hand, without understanding we are unable to design suitable power supply, (…)”.

c.       Page 2; line 52: “(…) hundreds of Volts and tens of Amps per (…)”
Units should not be written with a capital letter.

d.       Page 2; line 72: “ (…) and it can be easily eliminate if necessary.”
It should be “and it can be easily eliminateD if necessary.”

e.       In multiple instances in the text, the author misspelled the word “envelope” by writing it in the verb form (envelop).

f.        Page 3 & 4; lines 121 & 128. The use of sub- and superscripts in the manuscript is, in general, in order. The only examples of not following the routine was in the case of U_br, in the two quoted lines.

g.       Page 9; line 281. The author has written “us” meaning “microseconds”. Please use the corresponding Greek letter “µ”.

h.       Reference to Fig.9 precedes that of Fig.8, which should not happen. Consider changing the order of figures or edit text to follow this convention.

a-h) Corrected.

2)      In Fig.2. There seems to be a slight mistake in the notation of time (t). In the caption, there is an annotation “Symbols t and R have the same meaning as in Fig.1.” It is slightly misleading because in Fig.1. the author used a non-abbreviated description of the x-axis, namely, “time” and not “t”.

A2. Name of abscissa on Fig.1 was changed by “t” instead of “time”.

3)      As I’ve mentioned, I am not of an electronics/electrical engineering background. Therefore, it would be the most welcome to elaborate briefly in the text on the meaning of Fig.5, and what does this signal processing procedure correspond to concerning the optimal work of the power supply during PEO coating formation.

A3. Detailed description of fig.5 has been added.

4)      It feels wrong to have “4.1 Prototype details” subsection in “Results and discussion”. Please consider moving it to the previous section. Alternatively, you may find it more fitting to make it a separate section of an article altogether.

A4. Yes, this looks quite reasonable.

5)      Please clarify what is meant in the sentence (Page 10; line 308): “Moreover, regime designer allows one to include special diagnostic sequences as well as markers for external triggers that allow to synchronise outstanding oscilloscopes with particular process moment.”

A5. This means that in special cases, it may be useful to be able synchronise outstanding oscilloscope with particular time point within total pulse sequence. Say, if only pulse #3 is of our interest, we can put a special mark into regime right before pulse #3, which will produce high level on special output pin every time the process reaches this step of the sequence. This signal can trigger recording process of outstanding oscilloscope, camera etc.

6)      In Fig.10., please clarify:

a.       What is the meaning of Q_h and Q_h/Q_tot. This quantity is also brought up in Fig.13b.

A6a. Information added into figure caption.

b.       I am not sure what are the meanings behind legends of charts 2 and 3 (Current, Glow, Hydr; and InpU, up, down).

A6b. Charts 2 and 3 are shown only for instance of signal waveforms and X-Y loops.

7)      In Fig.11., please try to indicate that L corresponds to the light intensity of the MDs (at least I suspect that is the reason behind it). I would also like to know, is collecting this additional ADC data trigger some response from the power supply that modifies the output signal to the process? Simply put, do the light emission observations affect how the device runs the process?

A7. Signal of light emission (L) has been added into figure caption as well as in text.

No, this particular experiment includes only automatic adjustment of diagnostic pulse amplitude to previous pulse train maximum voltage. In order to reasonably include light emission into operational conditions it is necessary to have at least phenomenological model of the process. Currently, such model is not yet found. However, designed power supply can operate in this way.

8)      As a follow up to the previous question, I would like to know, because it is not immediately clear from the article: does the result of the diagnostic pulse affect the process conditions in the subsequent pulse train?

A8a. No. In that particular experiment the main idea was to add a small perturbation that will not affect the main process, only for diagnostic purposes in respect of relatively well known process. As I have written above, in order to close loop including external parameters the model of the process is required. It is necessary to write down a function, which connects particular response (e.g. light intensity) and output process parameters (e.g. current or voltage limit).

Q8b. It would be valuable to use such a diagnostic pulse for the sake of studying the process or the reproducibility of the desired quality of the coatings. However, being able to tune the power supply to respond specifically to the given diagnostic pulse response would be an outstanding step forward in the commercialization of PEO in industry.

Yes, this is the main goal of this design. Section 4 was complimented by following statement to clearly stress the idea: “Finally, advanced control method can be employed, when control function y = f(u,i) (see section 3.1) includes additional parameters (and/or conditions) derived from analysis of diagnostic pulse response, optical signal etc.”

Round  2

Reviewer 1 Report

Accept

Reviewer 3 Report

The author of the article did a marvelous job editing the manuscript to cope with the doubts that I have outlined in the previous round of the review. I would also like to express my thanks for addressing my questions. I feel that the article can be accepted in the present form. As a consequence, it will not lose its scientific merit. However, from the language point of view, it could still be improved. I leave it to the editors to decide whether to accept the article in the present version or to advise the author to make further corrections.